# Identification of Compound Heterozygous *EVC2* Gene Variants in Two Mexican Families with Ellis–van Creveld Syndrome

**DOI:** 10.3390/genes14040887

**Published:** 2023-04-09

**Authors:** Nancy Negrete-Torres, María del Carmen Chima-Galán, Ernesto Antonio Sierra-López, Janet Sánchez-Ramos, Isela Álvarez-González, Julia Reyes-Reali, María Isabel Mendoza-Ramos, Efraín Garrido-Guerrero, Dante Amato, Claudia Fabiola Méndez-Catalá, Glustein Pozo-Molina, Adolfo René Méndez-Cruz

**Affiliations:** 1Laboratorio de Genética y Oncología Molecular, Laboratorio 5, Edificio A4, Carrera de Médico Cirujano, Facultad de Estudios Superiores Iztacala, Universidad Nacional Autónoma de México, Tlalnepantla 54090, Mexico; nadeneto19@live.com (N.N.-T.); janetbiologia@gmail.com (J.S.-R.); danteamato1@gmail.com (D.A.); glustein@iztacala.unam.mx (G.P.-M.); 2Laboratorio de Genética, Escuela Nacional de Ciencias Biológicas Zacatenco, Instituto Politécnico Nacional, Ciudad de México 07738, Mexico; isela.alvarez@gmail.com; 3Centro Médico Nacional “20 de Noviembre”, ISSSTE, Ciudad de México 03229, Mexico; carmenchimag@yahoo.com.mx (M.d.C.C.-G.);; 4Laboratorio de Inmunología, Unidad de Morfofisiología y Función, Facultad de Estudios Superiores Iztacala, Universidad Nacional Autónoma de México, Tlalnepantla 54090, Mexico; reali@unam.mx (J.R.-R.); merisam06@iztacala.unam.mx (M.I.M.-R.); 5Departamento de Genética y Biología Molecular, CINVESTAV-IPN, Ciudad de México 07360, Mexico; 6División de Investigación y Posgrado, Facultad de Estudios Superiores Iztacala, Universidad Nacional Autónoma de México, Tlalnepantla 54090, Mexico

**Keywords:** *EVC2* gene, Ellis–van Creveld syndrome, whole exome sequencing, ciliopathies, Mexican patients

## Abstract

Background: Ellis–van Creveld syndrome (EvCS) is an autosomal recessive ciliopathy with a disproportionate short stature, polydactyly, dystrophic nails, oral defects, and cardiac anomalies. It is caused by pathogenic variants in the *EVC* or *EVC2* genes. To obtain further insight into the genetics of EvCS, we identified the genetic defect for the *EVC2* gene in two Mexican patients. Methods: Two Mexican families were enrolled in this study. Exome sequencing was applied in the probands to screen potential genetic variant(s), and then Sanger sequencing was used to identify the variant in the parents. Finally, a prediction of the three-dimensional structure of the mutant proteins was made. Results: One patient has a compound heterozygous *EVC2* mutation: a novel heterozygous variant c.519_519 + 1delinsT inherited from her mother, and a heterozygous variant c.2161delC (p.L721fs) inherited from her father. The second patient has a previously reported compound heterozygous *EVC2* mutation: nonsense mutation c.645G > A (p.W215*) in exon 5 inherited from her mother, and c.273dup (p.K92fs) in exon 2 inherited from her father. In both cases, the diagnostic was Ellis–van Creveld syndrome. Three-dimensional modeling of the *EVC2* protein showed that truncated proteins are produced in both patients due to the generation of premature stop codons. Conclusion: The identified novel heterozygous *EVC2* variants, c.2161delC and c.519_519 + 1delinsT, were responsible for the Ellis–van Creveld syndrome in one of the Mexican patients. In the second Mexican patient, we identified a compound heterozygous variant, c.645G > A and c.273dup, responsible for EvCS. The findings in this study extend the *EVC2* mutation spectrum and may provide new insights into the *EVC2* causation and diagnosis with implications for genetic counseling and clinical management.

## 1. Introduction

Ciliopathies are a group of rare genetic diseases that are characterized by defects in the biosynthesis and/or function of the cilia, causing a wide range of overlapping multisystemic clinical manifestations such as hydrocephalus, intellectual disability, retinal abnormalities, craniofacial defects, skeletal abnormalities, heart defects, kidney, pancreatic and liver cysts, sterility, lung and airway abnormalities, among others [1,2,3]. Ciliopathies can be subdivided into motile and non-motile ciliopathies [1,2]. Non-mobile, or sensory ciliary, disorders are defined as a heterogeneous group of genetic diseases caused by defects in the assembly or functioning of the 9 + 0 primary cilium, that present with superimposed clinical and radiographic characteristics that present with distinctive skeletal changes with ectodermal derived structures alterations in the teeth, nails, etc., growth defects and characteristic facial features, and may also be accompanied by retinal, nervous system, hepatic, cardiac, gastrointestinal and genitourinary manifestations [2,3,4,5].

These diseases are inherited as autosomal recessive traits, such as Ellis–van Creveld syndrome (EvCS) (OMIM # 225500), Sensenbrenner syndrome (OMIM # 218330), Jeune syndrome (OMIM # 208500), lethal polydactyly and short rib syndromes (OMIM # 263520), besides Weyers acrofacial dysostosis (WAD) (OMIM # 193530), which is inherited in an autosomal dominant pattern. Approximately 23 to 30 affected genes have been associated with these diseases [2,3,4,5,6].

Ellis–van Creveld syndrome, also called chondroectodermal dysplasia, was described in 1940 by Ellis and van Creveld [6,7,8,9]. It is defined as a skeletal dysplasia with autosomal recessive inheritance that presents as abnormal development in the ectodermal and mesodermal structures [6,7,10,11]. It is characterized by the presence of short stature, short limbs, brachydactyly, postaxial polydactyly, defects and absence of dental pieces, dysplastic nails, hair alterations, and cardiac malformations that occur in 60% of cases—with septal defects being the most frequent commonly presented [5,7,8,12]—and may also present with rib hypoplasia, and multiple lip-gingival adhesions; cognitive development is usually normal and the women are often infertile [1,6,13].

This syndrome is infrequent—its incidence varies depending on the population to be studied and in some cases it is unknown. A prevalence of 7 per 1,000,000 newborns, or 1.1 per 100,000 newborns, is reported [13]. There are populations with a higher number of cases, such as the Amish community in Lancaster Country, Pennsylvania, USA [10,13].

EvCS is caused by mutations in the *EVC* and *EVC2* genes (subunits 1 and 2 of the ciliary complex) located on chromosome 4p16.2, with 29 and 26 exons, respectively, in a 5′ to 5′ head-to-head configuration, and account for approximately 70% of the EvCS cases [14,15,16,17,18,19]. In addition, mutations in the WDR35 gene have been associated to clinical features similar to these two syndromes [8,15,19]. The *EVC*/*EVC2* genes encode transmembrane proteins that are located in the basal zone of the primary cilium and that play an important role in the activation and function of the Hedgehog pathway [20,21,22].

The Sonic Hedgehog (Shh) pathway plays an important role in the development of various organs, mainly craniofacial, in the ectoderm, notochord, and neural tube, and also has receptors in the mesenchyme for the constitution of the head, neuroectoderm, facial ectoderm, pharynx, and distal extremities [23,24,25,26,27]. In addition, Indian Hedgehog (Ihh) (a homologue of Shh) participates in bone and cartilage development and has redundant effects with the Shh pathway [24].

Shh binds to its target cells through the transmembrane receptor Patched1 (PTC1) generating the activation of the Smoothened protein (SMO), promoting its accumulation in the membrane of the tip of the primary cilium and the Shh/PTC1 are degraded in the proteasome; however, if the Shh/PTC1 junction is not generated, then the PTC1 suppresses the SMO release and accumulates in the ciliary membrane [23,24,26,27,28]. The SMO protein interacts to form a complex with the Ellis–van Creveld syndrome (*EVC*)/Limbin-(*EVC2*) proteins in the ciliary zone called the *EVC* zone, participating upstream of the Suppressor of fused (SuFu) regulator to promote the activation of the GLI1-3 transcription factors by dissociating them [24,28,29,30,31].

The aim of this study is to present the clinical cases of two Mexican patients presenting with clinical and molecular characteristics of Ellis van Creveld syndrome.

## 2. Cases and Methods

The complete clinical evaluation was carried out by clinical geneticists and the probable diagnosis was established. Later, the molecular diagnosis was done through whole exome sequencing (WES). Two families who attended a genetic consultation at the Centro Médico Nacional “20 de Noviembre” del ISSSTE were included. All the subjects that provided informed consent were eligible for participation. This study was conducted with the approval of the Institutional Review Board of Facultad de Estudios Superiores Iztacala (FESI). The clinical data for the study was extracted from the clinical history of each individual. Complying with the bioethical statements, an informed consent was given and all the necessary information about the study was provided to the patients. Genomic DNA was extracted from peripheral blood obtained from the affected individuals and their parents.

### 2.1. Case Report

#### 2.1.1. Case 1

The Ellis–van Creveld syndrome diagnosis was made using characteristic clinical manifestations. The proband (pedigree III:7) was a 14-year-old female, born to an unaffected couple (Figure 1a). A physical examination revealed a height of 132 cm (below 5 centiles, according to the tables of Centers of Disease Control and Prevention), a central and large nose with a regular nose bridge, and a broad-based bulbous tip (Figure 1b). Intraoral examination showed multiple oral frenula (upper and lower); dysplastic teeth and hypodontia; absence of two maxillary incisors and the bilateral presence of the first premolar and first molar; mandibular anodontia with the bilateral presence of canines, first premolar, and first molar (Figure 1c,d). Likewise, a chest murmur in aortic focus of systolic type. Additionally, wide hands with wide fingers, brachydactyly, and nail hypoplasia (Figure 1e); and a bilateral postaxial polydactyly resection scar (Figure 1f). Moreover, bilateral genu valgum and short toes with nail hypoplasia were revealed (Figure 1g). No congenital heart defect was found (normal echocardiogram and electrocardiogram). All relatives were unaffected with no similar features; however, the patient had a brother (pedigree III:8) with coarctation of the aorta, who died at 9 months of age.

#### 2.1.2. Case 2

The proband (pedigree III:11) was a 15-year-old female, born to an unaffected couple (Figure 2a). Physical examination revealed a height of 145 cm (below 5 centiles, according to the tables of Centers of Disease Control and Prevention), a normal nasal bridge, a straight nasal septum, a bulbous nasal tip, a wide columella, and a short philtrum (Figure 2b). Additionally, an oral cavity with an intact palate, a central uvula, and hypodontia were found (Figure 2c,d). Pectus carinatum and rhythmic heart sounds without aggregated sounds were revealed; her electrocardiogram and echocardiogram were normal. In the upper limbs, she presented brachydactyly, bilateral dysplastic thumb nails, wide fingers, a scar in the ulnar region of both hands, and cubitus valgus (Figure 2e,f). In the lower limbs, there were dysplastic nails, bilateral increased space between the first and second toe, bilateral second long toe, and genu valgum (Figure 2g). All relatives were unaffected with no similar features; however, the patient’s parents had a previous miscarriage (pedigree III:10). Moreover, in the first mother’s relationship, there was a history of two miscarriages (pedigree III:13 and III:14).

### 2.2. Molecular Genetic Studies

#### 2.2.1. DNA Isolation

Genomic DNA was extracted from peripheral blood samples using the DNeasy Blood & Tissue Kit (QIAGEN, Darmstadt, Germany). The integrity of the dsDNA was analyzed by electrophoresis on 0.8% agarose gels stained with the non-toxic Sybr Green dye. The purity of the dsDNA extraction was obtained with nanospectrometry (Implen) with the 260/280 nm ratio > 1.8 and <2.2. The DNA was quantified by fluorometry with the QuantiFluor dsDNA kit and the Quantus fluorometer (Promega, Madison, WI, USA).

#### 2.2.2. Whole Exome Sequencing

Genomic DNA samples were sent to Macrogen Inc. Service (Macrogen, Seoul, Korea) for WES. DNA was fragmented and enriched for exome sequences using the Sure Select Human All Exon V6 kit (Agilent Technologies, Santa Clara, California, USA) according to the manufacture’s protocol. The kit had a target size of 60 Mb and covered 99% of the most relevant databases, such as *RefSeq* and *OMIM_cds*, and was optimized to obtain data with high uniform coverage (>80%) at 100× depth. Libraries were sequenced on an Illumina NovaSeq 6000 platform (Illumina San Diego, CA, USA).

#### 2.2.3. READ Mapping and Variant Analysis

Sequencing data were first analyzed by determining the quality of the raw readings with the FastQC v0.11.9 software (Babraham Institute, Cambridge, UK), which determines the Phred quality score. The selected reads to further analyze showed a Phred score ≥ 30.

Subsequently, the optimal reads were aligned to the human reference genome, GRCh38, in a pair-end mode using the Burrows–Wheeler Aligner (BWA) tool [32]. Then, it is ordered by coordinates and indexing using the SAMtools v1.16.1 software [33]. After, recalibrations were carried out with the Genome Analysis Toolkit (GATK4) [34]. Once the reads were recalibrated, marked, and de-duplicated, we proceeded with the variant calling using the GATK4; then, we started with the identification of single nucleotide variants and proceeded with their recalibration by comparing them with the databases *dbSNP* [35], *ClinVar* [36], and *1000 Genomes* [37]; then, the identification and recalibration of the InDels was also carried out with the same databases. The FUNCONTATOR tool (part of GATK) was used to annotate the identified variants. Variants were further annotated using in-house exome databases with 60 Mexican controls and the *Exome Aggregation Consortium* (ExAC) “https://gnomad.broadinstitute.org/” (accessed on 24 September 2022) databases. Finally, a detailed analysis of the genes WDR35, *EVC,* and *EVC2* was conducted. Subsequently, an exhaustive search was made in the *ClinVar*, *OMIM*, *DisGeNET* (Integrative Biomedical Informatics Group GRIB/IMIM/UPF), and *MedGen* (National Center for Biotechnology Information) databases for the genotypic determination of possible mutations; finally, they were correlated with the clinical features of the patients to determine which was the possible cause of the syndromes presented.

To identify pathogenic single nucleotide variants (SNVs), each variant was evaluated based on the available information from the following databases: *HGMD* http://www.hgmd.cf.ac.uk (accessed on 24 September 2022), *ClinVar* “https://www.ncbi.nlm.nih.gov/clinvar” (accessed on 14 September 2022), *LSDBs* https://www.humanvariomeproject.org/ (accessed on 15 September 2022), *NHLBI Exome Sequencing Project* “https://evs.gs.washington.edu/EVS/” (accessed on 16 September 2022), *1000 Genomes* “http://www.internationalgenome.org” (accessed on 24 September 2022), and *dbSNP* “https://www.ncbi.nlm.nih.gov/snp/”(accessed on 26 September 2022). Published literature, clinical correlation and its predicted functional or splicing impact using evolutionary conservation analysis and computational tools such as AlignGVGD “http://agvgd.hci.utah.edu/” (accessed on 24 October 2022), MAPP “http://mendel.stanford.edu/SidowLab/downloads/MAPP” (accessed on 24 September 2022), MutationTaster, 8 PolyPhen-2,9 SIFT10 and SNAP “https://rostlab.org/services/snap2web” (accessed on 24 September 2022) were also used. We excluded sequence variants with low allele frequencies (>0.05).

Once the probable mutation was determined, a graphic display was made with the Integrative Genomics Viewer (IGV) software [38]. The allelic frequency of the variant found was consulted in the *gnomAD* database v3.1.2 [39]. We also used the *VarSome* platform [40], an impact analysis tool for genetic variants, and the in silico predictors REVEL [41], CADD [42], SpliceAI [43] and PrimateAI [44]. The variants were classified according to the standards and guidelines for the interpretation of sequence variants of the American College of Medical Genetics and Genomics and the Association for Molecular Pathology [45,46].

#### 2.2.4. PCR Amplification and Sanger Sequencing

Family segregation studies and variant validation analyses were performed by Sanger sequencing. Exons 2, 4, 5, and 14 of the *EVC2* gene were amplified from genomic DNA by conventional PCR. The sequences of the primers used to amplify these exons were:
5′AGAATGGCGTGAACCTAGGA3′ and 5′TCCACTGTGCACTAACGCTT3′ for exon 2;5′CTGGTAAGCACACGGTACAT 3′ and 5′TTGAAAACTGTCAGGTACCCT 3′ for exon 4;5′TGATAAATTCCCAGGCCCTC3′ and 5′CCACTGTGAGGATTAGGAGA3′ for exon 5;5′GAGATTGTTGGGGAAAAGGC3′ and 5′GGCACTCACATGAAGATCAG3′ for exon 14.


PCR products were purified using the AMPure XP purification kit (Beckman Coulter, IN, USA), followed by Sanger sequencing [47,48].

#### 2.2.5. 3D-Modeling

A prediction of the three-dimensional structure of the protein was made starting with the coding sequence of the *EVC2* gene that was taken from the NCBI (National Center for Biotechnology Information) (National Center for Biotechnology Information, 2022). Once the nucleotide position of the mutations was identified into the wild-type gene sequence, the modified gene sequences were generated. We used the Biomodel Transcription and Translation Web Tool “https://biomodel.uah.es/en/lab/cybertory/analysis” (accessed on 8 November 2022); then, the mutated *EVC2* sequence was transcribed to mRNA, and subsequently translated to the amino acid sequence. To generate the tertiary structure of the *EVC2* mutants proteins, their amino acid sequence was computed by the SWISS-MODEL server homology modelling pipeline [49] https://swissmodel.expasy.org/assess (accessed on 8 November 2022), which relies on ProMod3 [50], a comparative modelling engine based on OpenStructure [51]. The *EVC2* template was obtained from the repository of protein structures AlphaFold [52] (Q86UK5) “https://alphafold.ebi.ac.uk/” (accessed on 9 November 2022), which was obtained through a bioinformatic prediction from the amino acid sequence of the *EVC2* protein, which does not have records in the RCSB Protein Data Bank (RCSB PDB) “https://www.rcsb.org/”(accessed on 9 November 2022). Subsequently, a modeling of the protein was carried out on the SWISS-MODEL website with the option of using a template; the amino acid sequence of the variant proteins was entered and the three-dimensional structures of each one were obtained. The prediction of the three proteins was visualized with the UCSF Chimera program [53] “https://www.cgl.ucsf.edu/chimera/” (accessed on 9 November 2022).

## 3. Results

The quality of the raw readings was analyzed using the FastQC v0.11.9 software, which showed a Phred quality score of ≥30 for all reads in both patients. After WES, reads were aligned to the human genome sequence assembly GRCh38. Variants were filtered to identify the most potential candidate variants.

We found two heterozygous variants in the proband of case 1. A novel heterozygote variant was detected in exon 4, a 2-nucleotides deletion (GG), and a 1-nucleotide insertion (T), c.519_519 + 1delinsT (electropherogram III:7) (Figure 3a). This sequence change affects a donor splice site in intron 4 of the *EVC2* gene, expecting to disrupt the RNA splicing. A second heterozygote variant in exon 14 of the *EVC2* gene was identified (electropherogram III:7) (Figure 3b), c.2161delC, which resulted in a frameshift of the protein at leu721 (p.L721fs). These compound heterozygote variants in the *EVC2* gene are associated with Ellis–van Creveld syndrome, which is consistent with the clinical findings of the patient. To confirm the zygosity of the variant, segregation analysis was performed. The results showed that the c.2161delC variant was inherited from the father (electropherogram II:7) (Figure 3b), while the c.519_519 + 1delinsT variant was inherited from the mother (electropherogram II:8) (Figure 3a). In accordance with the guidelines of the American College of Medical Genetics and Genomics (ACMG), c.2161delC is a null variant (frameshift) for which loss of function is a known mechanism associated with EvCS; and as a result, it was determined to be pathogenic (1PVS, 2PM). c.519_519 + 1delinsT is a splice site donor variant generating a null variant in the *EVC2* gene, and as a result, it is likely-pathogenic (PVS1 and PM2).

In the family history, there was no record of a similar disorder that had been aborted or had been born. This suggests that these mutations have recently occurred in the family, and they are associated with an autosomal recessive inheritance.

Each variant was only found in one of the parents, respectively; furthermore, the c.519_519 + 1delinsT variant was not detected in either in-house exome databases, and it has not been reported in the *ClinVar*, *gnomAD*, or *ExAC* databases. The c.2161delC variant is considered as pathogenic and has been previously reported in *ClinVar*.

Both variants generate a truncated *EVC2* protein as can be seen by the generation of a three-dimensional model. The analysis predicts the production of both a 173 and a 738 amino acids protein encompassing only a part of its coiled coil domain (Figure 3c,d); that is, the Weyers peptide is not translated, which prevents the formation of the SMO/*EVC*/*EVC2* complex, so the Shh signaling pathway cannot be activated.

The patient of case 2 (electropherogram III:11) was a compound heterozygote with a 1-nucleotide insertion in exon 2 inherited from her father (electropherogram II:11), c.273dup (c.273_274insT as reported), resulting in a frameshift of the *EVC2* protein, p.K92fs (Figure 4a), and a nonsense mutation in exon 5 in the *EVC2* gene inherited from her mother (electropherogram II:12), c.645G > A (p.W215*) (Figure 4b). Both mutations are considered as pathogenic and have been previously reported in *ClinVar*, in *Leiden Open Variation Database* (LOVD3) https://databases.lovd.nl/shared/variants/0000710769#00007281 (accessed on 15 December 2022), and by Tompson et al. (2007) [18].

The evaluation of the pathogenicity of the c.645G > A and c.273dup variants showed that, in accordance with the guidelines of the ACMG, are null variants in the *EVC2* gene for which the loss of function is a known mechanism associated with EvCS, and as a result, these are pathogenic variants (1PVS, 2PM, 5PP).

In the family history, it seems that all relatives were clinically unaffected; however, there is a history of three miscarriages. Both variants generated a truncated *EVC2* protein, as can be shown by the generation of the three-dimensional model (Figure 4c,d). The in silico analysis predicts the production of both a 92 and a 215 amino acids proteins, respectively. The p.K92fs variant induces a change in the reading frame, generating a 91 amino acid peptide, which corresponds only to the translation of the signal peptide; therefore, the protein cannot bind to the cilium membrane to form the SMO/*EVC*/*EVC2*. The variant p.W215* that encodes another premature stop codon induces the synthesis of only the signal peptide domain, the β sandwich domain and a segment of the transmembrane domain, which could cause the *EVC2* protein not to bind to the membrane of the primary cilium, and thus prevents the Shh pathway activation (Figure 4c,d).

## 4. Discussion

Ciliopathies are caused by defects in the cilia structure or function. In EvCS, the structure of the cilia is normal, but the Hh and Fibroblast growth factor (FGF) signaling pathways are impaired [21,27]. Reduced Hedgehog signaling and increased FGF signaling at the growth plaque was reported in *EVC2* mutant mice [54]. The Hehgehog signaling starts the association of *EVC2* with Smoothened (Smo). Smo-*EVC2* signaling complex at the *EVC* zone is essential for Hh signal transmission [20,28].

In the present study, we identified one novel heterozygous variant in the *EVC2* gene, c.519_519 + 1delinsT, in patient 1. The mutations c.2161delC (p.L721fs) (patient 1), and c.645G > A (p.W215*) and c.273dup from patient 2, were previously reported. These variants were detected in two Mexican families that allowed us to establish the diagnosis of Ellis–van Creveld syndrome. The four variants generate premature stop codons. In both cases, we confirmed that the compound heterozygous mutations of the patients were inherited from their non-affected parents, respectively, consistent with recessive inheritance.

These patients showed short stature, bilateral hands postaxial polydactyly, and ectodermal anomalies involving nails and teeth. According to the guidelines of the American College of Medical Genetics and Genomics, these variants are considered pathogenic and/or likely-pathogenic, so it could explain the clinical manifestations observed in these patients. EvCS is a skeletal dysplasia characterized by short limbs, short ribs, postaxial polydactyly, and dysplastic nails and teeth, which could also be accompanied by congenital cardiac defects—most commonly, a defect of primary atrial septation [14]. Both patients manifested typical clinical features, but congenital heart defects were not observed in the electrocardiograms and echocardiograms.

Moreover, there are conflicting reports suggesting the existence of genetic heterogeneity in EvCS. Tompson et al. (2007) identified in a study with 65 affected individuals, that there were *EVC*/*EVC2* mutations in only 45/65 cases (69%) [18]. In another panel of 32 families with a clinical diagnosis of *EVC* and Weyers syndromes, 27/32 cases of *EVC* and 2/2 cases of Weyers have *EVC*/*EVC2* mutations [55]. Both studies suggested the possibility of genetic heterogeneity of additional genes involved in *EVC* and Weyers syndromes. However, in a cohort of 36 *EVC* and Weyers families, pathogenic changes were identified in all cases, providing no support of heterogeneity [56]. Furthermore, in our study, both families showed pathogenic variants in the *EVC2* gene. Therefore, additional studies are needed to clarify the *EVC*/*EVC2* mutation spectrum and heterogeneity.

To date, according to the suite of bioinformatics tools for processing and annotation of NGS data, *VARSOME* https://varsome.com/gene/hg38/evc2 (accessed on 29 November 2022), more than 450 mutations in *EVC2* have been reported, of which 224 are pathogenic or likely-pathogenic variants, and 250 variants of uncertain significance (VUS) have been associated to patients with *EVC* and Weyers syndromes (summarized in Table 1).

Homozygous or compound heterozygous mutations in the *EVC* and *EVC2* genes can cause EvCS [14]. The phenotype associated with variants in either of these genes is indistinguishable because *EVC* and *EVC2* act in a common pathway [18]. These genes are arranged in a head-to-head configuration that is conserved from fish-to-man [57]. These two proteins interact directly to form a complex at the primary cilium membrane (*EVC* zone), which is essential for normal endochondral growth and intramembranous ossification [54]. In vitro studies have shown that Hedgehog (Hh) signal transduction was defective in cells lacking *EVC*, thus suggesting that it is essential for Indian Hedgehog (Ihh) signaling in the cartilage growth plate [18]. Blair et al. (2011) showed that the presence of *EVC* and *EVC2* at the basal body and cilia membrane was co-dependent, and that *EVC2* is a positive regulator of the Hh signaling pathway [20].

Although great achievements have been made, including the generation of animal models that explain the consequences of mutations in these genes, the genotype–phenotype correlations in Ellis–van Creveld are still unclear. For example, cardiovascular malformations only occur in 60% of patients. However, the affected girls in both families in our study are free of cardiovascular disturbance. Therefore, more intensive studies are necessary to further identify correlations between the genotypes and phenotypes. Until now, a considerable number of *EVC* and *EVC2* mutations have already been identified in patients with EvCS [18]. The majority of the variants reported are nonsense or frameshift mutations, similar to what we observed in these two Mexican families, that introduce a stop codon and, therefore, it is likely that transcripts undergo nonsense-mediated decay [8,18]. We identified four variants; furthermore, one had not been previously reported (c.519_519 + 1delinsT). These mutations generate the production of truncated *EVC2* proteins, as predicted by 3D-modeling. The c.2161delC and c.273dup variants are frameshift mutations that generate premature stop codons. The c.519_519 + 1delinsT variant affects a donor splice site in intron 4. The result using the in silico predictor SpliceAI is 1, which means that this variant that disrupts the donor or acceptor splice site typically leads to a loss of protein function. Disruption of this splice site has been observed in individuals with *EVC2*-related conditions (PMID: 17024374; Invitae). Splice donor and acceptor sites are highly conserved during evolution. It is known that abnormal splicing can cause human genetic diseases. The most common consequence of splicing mutations usually leads to aberrant pre-mRNA splicing, which results in exon skipping, activation of cryptic splice sites, creation of pseudo-exons within introns, or intron retention [58]. Interestingly, the splicing site mutation identified in our EvCS patient 1 could result in the appearance of a premature stop codon within the sequence of intron 4, generating a hypothetical 173 amino acid truncated protein, as predicted by the 3D-protein modeling.

Concerning the families of the patients, the recurrence of risk is 25%. Although the patients had not been diagnosed until adolescence, the fetus of any consequent pregnancy can be diagnosed prenatally according to the existence of both clinical features of this syndrome, and the knowledge of their genetic background.

## 5. Conclusions

In summary, we identified one novel (non-reported) mutation in the *EVC2* gene in one Mexican family, c.519_519 + 1delinsT, and three previously reported variants, c.2161delC (p.L721fs), c.645G > A (p.W215*), and c.273dup. Both patients present compound heterozygous mutations. The findings could contribute to a further understanding of the relationship between phenotypes and genotypes in the *EVC* syndrome. Current findings further expand the EvCS mutational spectrum and argue for wider genetic heterogeneity in EvCS. Further understanding of the pathogenesis of EvCS will be needed by analyzing more families.

## 6. Web Sources

HGMD: ”http://www.hgmd.cf.ac.uk” (accessed on 15 December 2022),ClinVar: “https://www.ncbi.nlm.nih.gov/clinvar/?term=evc2%5Bgene%5D&redir=gene” (accessed on 15 December 2022), LSDBs: “https://www.humanvariomeproject.org/” (accessed on 15 December 2022),NHLBI Exome Sequencing Project: “https://evs.gs.washington.edu/EVS/” (accessed on 15 December 2022),1000 Genomes: “http://www.internationalgenome.org” (accessed on 15 December 2022),dbSNP: “https://www.ncbi.nlm.nih.gov/snp/” (accessed on 15 December 2022),AlignGVGD: “http://agvgd.hci.utah.edu/” (accessed on 15 December 2022),MAPP: “http://mendel.stanford.edu/SidowLab/downloads/MAPP” (accessed on 15 December 2022),SNAP: “https://rostlab.org/services/snap2web” (accessed on 15 December 2022),LOVD3: “https://databases.lovd.nl/shared/variants/0000710769#00007281” (accessed on 28 January 2023),VARSOME: “https://varsome.com/gene/hg38/evc2” (accessed on 15 December 2022).

## Figures and Tables

**Figure 1 genes-14-00887-f001:**
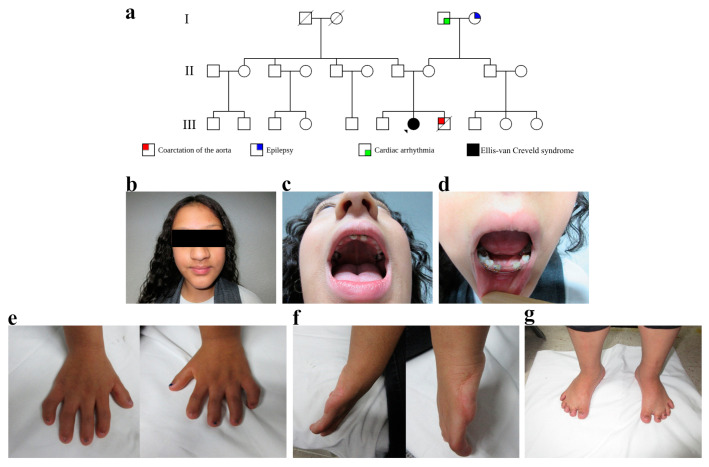
Clinical features of a patient with Ellis–van Creveld syndrome. (**a**) Pedigree of a Mexican family with compound heterozygote *EVC2* variants. The square denotes a male family member; the circle presents a female family member; the slashed symbol indicates a deceased family member; the fully shaded symbol shows the patient with EvCS (III:7); III:8 shows a patient with coarctation of the aorta. (**b**) Central and large nose with a regular nose bridge and a broad-based bulbous tip. (**c**,**d**) Multiple oral frenula, dysplastic teeth, and hypodontia; absence of two maxillary incisors and the bilateral presence of the first premolar and first molar, mandibular anodontia with bilateral presence of canines, and the first premolar and first molar. (**e**) Wide hands with wide fingers, brachydactyly, and dysplastic nails. (**f**) Bilateral postaxial polydactyly resection scar. (**g**) genu valgum and short toes with nail dysplasia.

**Figure 2 genes-14-00887-f002:**
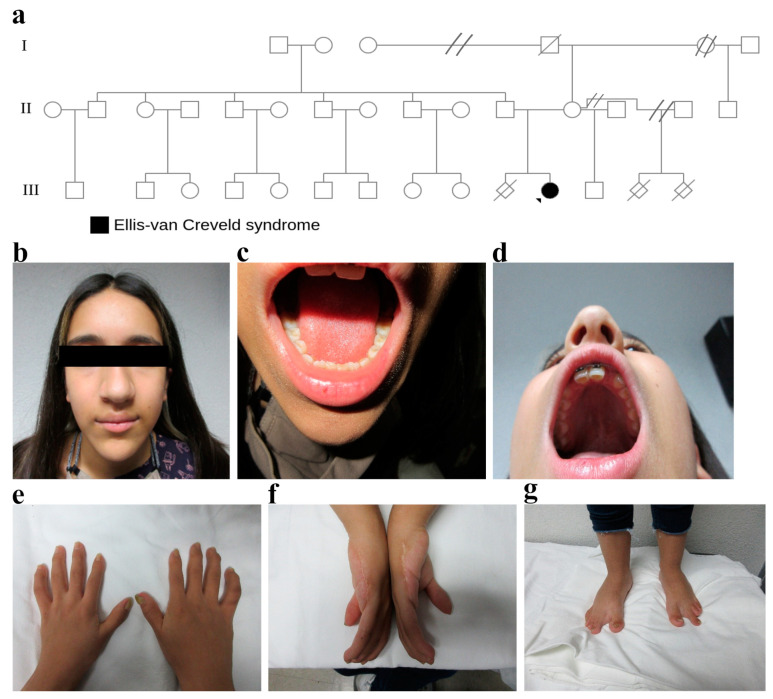
Clinical features of a patient with EvCS. (**a**) Pedigree of a Mexican family with an autosomal recessive *EVC2* variant. The fully shaded symbol shows the patient with EvCS (III:11). (**b**) A normal nasal bridge, a straight nasal septum, a bulbous nasal tip, a wide columella, and a short philtrum. (**c**,**d**) An oral cavity with an intact palate, a central uvula, and hypodontia. (**e**) Brachydactyly, bilateral dysplastic thumb nails, and wide fingers. (**f**) A resection scar in the ulnar region of both hands. (**g**) In the lower limbs—dysplastic nails, bilateral increased space between the first and second toes, a bilateral second long toe, and genu valgum.

**Figure 3 genes-14-00887-f003:**
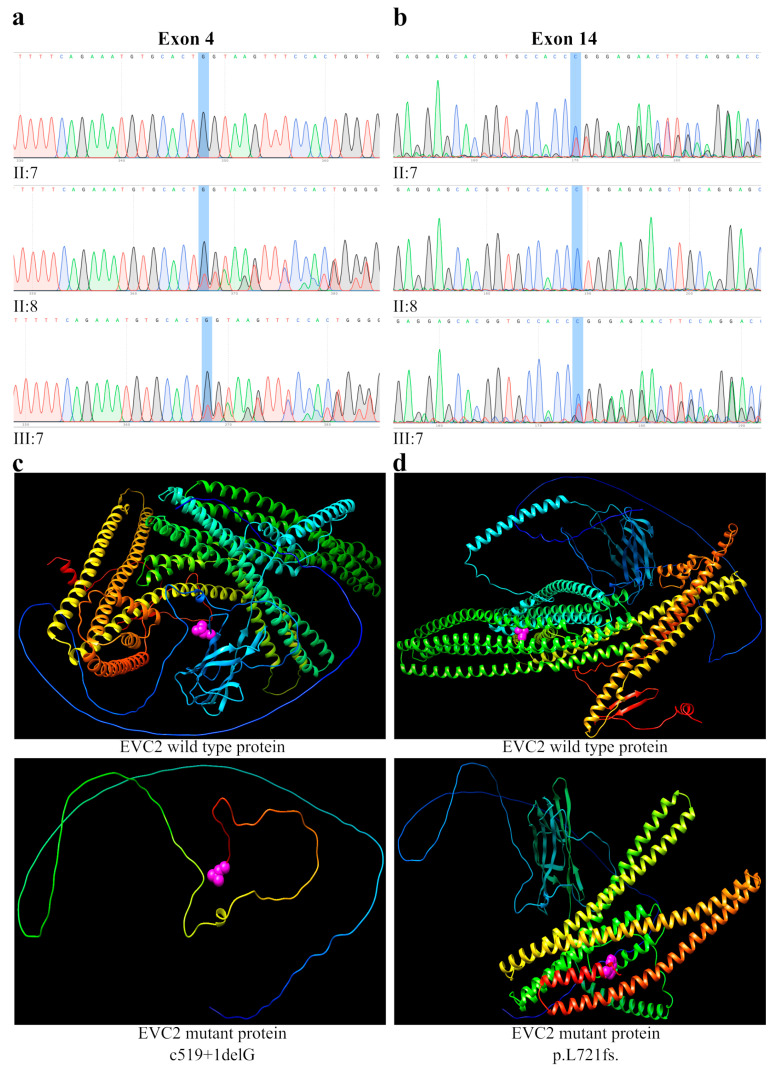
The results of the genetic analysis in the *EVC2* gene in a pedigree with EvCS. (**a**) The results of Sanger sequencing of exon4, the patient (electropherogram III:7) and her mother (electropherogram II:8) share a heterozygous donor splice site variant of *EVC2*: c.519_519 + 1delinsT. The father (electropherogram II:7) shows a normal *EVC2* sequence. (**b**) Sanger sequencing of exon 14; the patient (electropherogram III:7) and her father (electropherogram II:7) share a heterozygous frameshift variant of *EVC2*: c.2161delC, p.L721fs. The mother (electropherogram II:8) shows a normal sequence. (**c**) Three-dimensional molecular model of the wild type *EVC2* protein (upper panel) showing the last encoded amino acid by exon 4 coding sequence in pink bubbles; the bottom panel shows the predicted truncated protein generated by translation from exon1 to exon 4, and the pink bubbles indicates the last amino acid. (**d**) Three-dimensional molecular model of the wildtype *EVC2* protein (upper panel) and the predicted protein (bottom panel) with heterozygous p.L721fs variant which generates a truncated protein, and the last amino acid is indicated in pink bubbles.

**Figure 4 genes-14-00887-f004:**
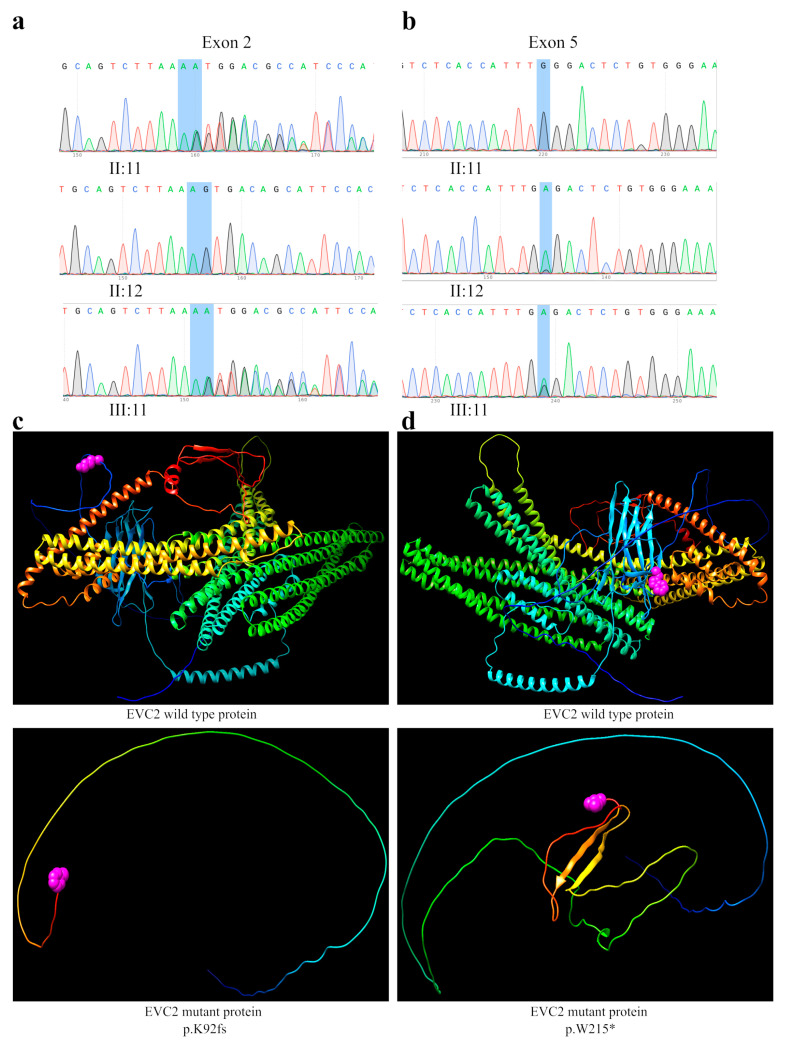
The results of the genetic analysis in the *EVC2* gene in a pedigree with EvCS. (**a**) The results of Sanger sequencing. The patient (electropherogram III:11) and his father (electropherogram II:11) share a heterozygous frameshift variant of *EVC2*—c.273dup, and p.K92fs in exon 2. The mother (electropherogram II:12) shows a normal *EVC2* sequence. (**b**) The results of Sanger sequencing. The patient (electropherogram III:11) and his mother (electropherogram II:12) share a heterozygous nonsense variant of *EVC2*: c.645G > A, p.W215* in exon 5. The father (electropherogram II:11) shows a normal *EVC2* sequence. (**c**) Three-dimensional molecular models of the wild type *EVC2* protein (upper panel), the *EVC2* protein with heterozygous p.K92fs variant (bottom panel), the mutant *EVC2* protein that generates truncated proteins; the pink bubbles indicate the last amino acid encoded. (**d**) Three-dimensional models of the wild type *EVC2* protein (upper panel), and the *EVC2* truncated protein with the heterozygous p.W215* variant (bottom panel); the pink bubbles indicate the last amino acid encoded.

**Table 1 genes-14-00887-t001:** Genetic variants detected in the *EVC2* gene with known pathogenicity.

Total Classified Variants (UniProt, ClinVar, VarSome & PubMed)						
*n* = 1021						
	Pathogenic		Uncertain Significance	Benign		
	*n* = 224		*n* = 250	*n* = 547		
Coding Impact	Pathogenic	Likely Pathogenic	Uncertain Significance (VUS)	Likely Benign	Benign	Total
**Synonymous**	0	0	29	389	24	442
**Missense**	5	2	188	10	27	232
**Nonsense**	63	18	2	0	0	83
**Frameshift**	50	41	3	0	0	94
**Inframe Indel**	0	1	13	0	0	14
**Splice junction loss**	7	37	0	0	0	44
**Non-coding**	0	0	15	82	15	112
**Total**	125	99	250	481	66	1021

## Data Availability

The data sets generated and/or analyzed during the current study are available from the corresponding author on reasonable request.

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
