# Peer review of "Identification of Compound Heterozygous EVC2 Gene Variants in Two Mexican Families with Ellis–van Creveld Syndrome"

_genes, 2023, doi:10.3390/genes14040887_

Round 1

Reviewer 1 Report

The authors describe two cases of Ellis-van Creveld syndrome and Weyers acrofacial dysostosis, an interesting topic if considering the rarity of the diseases. Despite this, there are major issues with the methodology and data presented.

The authors stated that that the novel c.2161delC heterozygous variant is responsible for Weyers acrofacial dysostosis (autosomal inheritance) in a patient previously diagnosed as EvCS (recessive inheritance), but it is not clear whether all the EVC2 coding regions have been analyzed by WES. Having detected the first variant (in a supposed recessive disease), why didn’t you look for a second variant analyzing the whole gene by Sanger sequencing?  The patient should also be tested by qRT-PCR for the presence of large deletions and duplications, already detected in previous studies [D’Asdia MC et al 2013]. Without these complete analyses, I don’t think it’s correct to associate the new variant with WAD, especially considering the presence of the same variant in the healthy father (Has incomplete penetrance ever been described?), as well as that all WAD-associated EVC2 variants have to date been restricted to exon 22 of the gene.   

The variant c.273_274insT defined as a novel variant should be described as c.273dup; in addition, it has been already reported in both LOVD and HGMD database, as well as in a previous publication also cited in the references [Tompson (2007), Hum Genet].

There is no information about technical/quality results of the WES analyses. What about the samples’ coverage? What about the EVC/EVC2 coverage? Have all the coding regions been analyzed?

In addition, it is not clear the requirement of three-dimensional modeling to show the truncated protein, already explained by the mutation types.

Considering the English language and style, I recommend a major revision, also considering high redundancy of many information, often unnecessary or inserted in the wrong sections.

Overall, I suggest to completely revised the paper, making it a clinical case presentation but only after performing all the additional molecular analyses previously described to exclude the presence of a second EVC2 variant. According to the family tree and to the first clinical suggestion, it is more reasonable thinking about a recessive disease.

Author Response

Reviewer #1 Specific comments:

The authors describe two cases of Ellis-van Creveld syndrome and Weyers acrofacial dysostosis, an interesting topic if considering the rarity of the diseases. Despite this, there are major issues with the methodology and data presented.

1. The authors stated that that the novel c.2161delC heterozygous variant is responsible for Weyers acrofacial dysostosis (autosomal inheritance) in a patient previously diagnosed as EvCS (recessive inheritance), but it is not clear whether all the EVC2 coding regions have been analyzed by WES. Having detected the first variant (in a supposed recessive disease), why didn’t you look for a second variant analyzing the whole gene by Sanger sequencing?  The patient should also be tested by qRT-PCR for the presence of large deletions and duplications, already detected in previous studies [D’Asdia MC et al 2013]. Without these complete analyses, I don’t think it’s correct to associate the new variant with WAD, especially considering the presence of the same variant in the healthy father (Has incomplete penetrance ever been described?), as well as that all WAD-associated EVC2 variants have to date been restricted to exon 22 of the gene.   

According to this observation, we reanalyzed the WES results and we found that patient 1 had another variant in exon 4 in the donor splicing site, this mutation was inherited by her mother. The mutation was confirmed by Sanger sequencing. The reanalyzed results show us that the correct diagnosis is Ellis-van Creveld syndrome. The patient inherited one mutation from her mother and one for her father, respectively. The Sanger sequencing for exon 22 did not show any mutation.

Finally, we made all the necessary changes in all the sections of the paper according to the new results, starting for the paper title: “Identification of novel EVC2 variants in two Mexican families with Ellis-van Creveld syndrome”.

2. The variant c.273_274insT defined as a novel variant should be described as c.273dup; in addition, it has been already reported in both LOVD and HGMD database, as well as in a previous publication also cited in the references [Tompson (2007), Hum Genet].

In reference to this observation, we reviewed the suggested databases and found that this mutation had been already reported. Additionally, we corrected the variant's nomenclature in all sections where it appears. The variant is described as c.273dup

3. There is no information about technical/quality results of the WES analyses. What about the samples’ coverage? What about the EVC/EVC2 coverage? Have all the coding regions been analyzed?

According to this observation, it was included in Results section the next paragraph lines 223-228:

The fragmented DNA for WES was enriched for exome sequences using the Sure Select Human All Exon V6 kit (Agilent Technologies). The kit has a target size of 60 Mb and covered 99% of the most relevant databases such as RefSeq and OMIM_cds and was optimized to obtain data with high uniform coverage (>80%) at 100X depth. The quality of the raw readings was analyzed using the FastQC v0.11.9 software, which showed a Phred quality score ≥ 30 for all reads

4. In addition, it is not clear the requirement of three-dimensional modeling to show the truncated protein, already explained by the mutation types.

In reference to this observation, We consider that this modeling is important to clarify the 3D structure. The 3D modeling gives a better idea of how affected the tertiary structure of the Evc2 protein is when we compare it against the wildtype, also, for potential readers who are not specialists in the area, it could clarify in a better way how these proteins are affected by the variants detected in patients.

5. Considering the English language and style, I recommend a major revision, also considering high redundancy of many information, often unnecessary or inserted in the wrong sections.

According to this observation, we did major changes in results and discussion sections.

6. Overall, I suggest to completely revised the paper, making it a clinical case presentation but only after performing all the additional molecular analyses previously described to exclude the presence of a second EVC2 variant. According to the family tree and to the first clinical suggestion, it is more reasonable thinking about a recessive disease.

In reference to this observation, we made all the corrections and changes sugested by reviewer.

Reviewer 2 Report

The manuscript Identification of novel EVC2 variants in two Mexican families with Weyers acrofacial dysostosis and Ellis-van Creveld syndrome by Nancy Negrete-Torres , María del Carmen Chima-Galán , Ernesto Antonio Sierra-López , Janet Sanchez-Ramos , Isela Álvarez-González , Julia Reyes‐Reali , María Isabel Mendoza‐Ramos , Efraín Garrido , Dante Amato , Claudia Fabiola Méndez-Catalá , Glustein Pozo-Molina , Adolfo René Méndez‐Cruz

describes new variants of Weyers acrofacial dysostosis (WAD) and Ellis-van Creveld syndrome (EvCS) in two patients and their family.

I read the text with a great interest. Introduction, appropriate methods and interesting results. One of the limitations is the low number of patients, but I understand the difficulties.

The manuscript is written in a clear way and cites relevant literature, the text is original, the schemes in Figures 2 and 4 are great. 

I definitely recommend this paper for publication in genes. The research is significant.

However I do have some minor comments:

Please, delete subheadings from the Abstract section.

In the Introduction section a scheme summarizing the symptoms of both Syndromes and mutations causing these syndromes. It would strengthen your manuscript.

Figures should be cited in the main text as (Figure 1 A) etc. not (FIG) Please check it through the text.

line 73- brackets are missing: is reported [13]

Subsections in the M&M and Results sections should be in italics.

Figure 1 and Figure 3. Please improve it< make it more accurate. The letters A-C and D-F should be located on the same level (Use line tool in Photoshop). The spaces between photos should be smaller and the same size( Use Photoshop lines and scales to check it).

Figure 2 and Figure 4. Please, reorganize letters A-D in a more accurate way. Letters A, B and C in Figure 2 should be on the same vertical line, letters C and D should be on the same horizontal line. Letters A, B and D in Figure 4 should be on the same vertical line.

In all figures please check font size (it should be 9) and style (look it in the template).

EVC2 gene should be in italics and protein without italics, please check it through the text.

line 167: please correct the citation style for (Jaganathan et al., 2019).

Please be consistent with abbreviations^ in the Introduction you write SHH pathway (lines 93-98), in the Results section you use Shh (line 325). Please, be consistent. Check it through the text.

The Discussion should be reorganized and divided into subsections: 1)difficulties of diagnosis; 2) genotype/phenotype correlation; 3) comments on the data provided in this study, highlighting novelties.

Author Response

The manuscript Identification of novel EVC2 variants in two Mexican families with Weyers acrofacial dysostosis and Ellis-van Creveld syndrome by Nancy Negrete-Torres , María del Carmen Chima-Galán , Ernesto Antonio Sierra-López , Janet Sanchez-Ramos , Isela Álvarez-González , Julia Reyes‐Reali , María Isabel Mendoza‐Ramos , Efraín Garrido , Dante Amato , Claudia Fabiola Méndez-Catalá , Glustein Pozo-Molina , Adolfo René Méndez‐Cruz

describes new variants of Weyers acrofacial dysostosis (WAD) and Ellis-van Creveld syndrome (EvCS) in two patients and their family.

I read the text with a great interest. Introduction, appropriate methods and interesting results. One of the limitations is the low number of patients, but I understand the difficulties.

The manuscript is written in a clear way and cites relevant literature, the text is original, the schemes in Figures 2 and 4 are great. 

I definitely recommend this paper for publication in genes. The research is significant.

However I do have some minor comments:

1. Please, delete subheadings from the Abstract section.

According to this observation,we made the sugested corrections.

2. In the Introduction section a scheme summarizing the symptoms of both Syndromes and mutations causing these syndromes. It would strengthen your manuscript.

In reference to this observation, we made major changes suggested by the observations of one of the reviewers, so with the changes made we do not consider it necessary to do the requested table. The changes were originated by the new analysis and interpretation of the results, beginning with the change of the title: “Identification of novel EVC2 variants in two Mexican families with Ellis-van Creveld syndrome

3. Figures should be cited in the main text as (Figure 1 A) etc. not (FIG) Please check it through the text.

According to this observation, the suggested corrections were made.

4. line 73- brackets are missing: is reported [13]

According to this observation, the suggested correction was made.

5. Subsections in the M&M and Results sections should be in italics.

According to this observation, the suggested corrections were made.

6. Figure 1 and Figure 3. Please improve it< make it more accurate. The letters A-C and D-F should be located on the same level (Use line tool in Photoshop). The spaces between photos should be smaller and the same size( Use Photoshop lines and scales to check it).

Figure 2 and Figure 4. Please, reorganize letters A-D in a more accurate way. Letters A, B and C in Figure 2 should be on the same vertical line, letters C and D should be on the same horizontal line. Letters A, B and D in Figure 4 should be on the same vertical line.

According to this observation, the suggested corrections were made.

7. In all figures please check font size (it should be 9) and style (look it in the template).

According to this observation, the suggested corrections were made.

8. EVC2 gene should be in italics and protein without italics, please check it through the text.

According to this observation, the suggested corrections were made.

9. line 167: please correct the citation style for (Jaganathan et al., 2019).

According to this observation, the suggested corrections was made.

10. Please be consistent with abbreviations^ in the Introduction you write SHH pathway (lines 93-98), in the Results section you use Shh (line 325). Please, be consistent. Check it through the text.

According to this observation, the suggested corrections were made.

11. The Discussion should be reorganized and divided into subsections: 1)difficulties of diagnosis; 2) genotype/phenotype correlation; 3) comments on the data provided in this study, highlighting novelties.

According to this observation, we made major changes suggested by the observations of one of the reviewers. The changes were originated by the new analysis and interpretation of the results, beginning with the change of the title: “Identification of novel EVC2 variants in two Mexican families with Ellis-van Creveld syndrome”. Also we made some changes in all sections.

Reviewer 3 Report

Comments and Suggestions for Authors

Nancy et al. identified a novel heterozygous EVC2 variant causing Weyers acrofacial dysostosis in one patient and a compound heterozygous variant in EVC2 causing Ellis-van Creveld syndrome in the second patient.

·      Please enlist the prioritized variant after exome analysis in the supplementary table. Is there any potentially interested comp. het variants were identifiedied in patient 1? 

·      In case 1, the author mentioned the father is unaffected, what clinical tests were performed on the father for diagnosis?

·      In both probands, which tests were performed in the patient to exclude the heart defect?

·      The author claimed reduced penetrance in case 1. Previously, is there any evidence of reduced penetrance in the EVC2 causing Weyers acrofacial dysostosis? If yes, please mention it in the discussion.

·      The third discussion needs to improve and there should be some comparison or relationship of current findings in Case 1 with previous studies to support the results.  

·      Please check the frequency of the new variants in other databases as well e.g Bravo, all of us, GME.

·      Spacing between the words and paragraph needs to double check.

·      The nomenclature of mutation should follow the same format as HGSV.

·      In line 319 correct the spelling of the abbreviation ACMG.  

·      Figure 1 needs to modify, the description is not well formatted.

·      The protein modeling picture should be larger so can easily visualize.

·      The web links of databases that are added in the results section can be added after the conclusion under the heading of web sources.

Author Response

Nancy et al. identified a novel heterozygous EVC2 variant causing Weyers acrofacial dysostosis in one patient and a compound heterozygous variant in EVC2 causing Ellis-van Creveld syndrome in the second patient.

 1. Please enlist the prioritized variant after exome analysis in the supplementary table. Is there any potentially interested comp. het variants were identifiedied in patient 1? 

According to this observation, we reanalyzed the WES results and we found that patient 1 had another variant in exon 4 in the donor splicing site, this mutation was inherited by her mother. The mutation was confirmed by Sanger sequencing. The reanalyzed results show us that the correct diagnosis is Ellis-van Creveld syndrome. The patient inherited one mutation from her mother and one for her father, respectively. The Sanger sequencing for exon 22 did not show any mutation.Finally, we made all the necessary changes in all the sections of the paper according to the new results, starting for the paper title: “Identification of novel EVC2 variants in two Mexican families with Ellis-van Creveld syndrome”.

2. In case 1, the author mentioned the father is unaffected, what clinical tests were performed on the father for diagnosis?

According to this observation, the reanalyzed results show us that the correct diagnosis is Ellis-van Creveld syndrome. The patient is a compound heterozygote. Her father is a carrier. Moreover, the clinical and physical evaluation did not show any disturbance.

3. In both probands, which tests were performed in the patient to exclude the heart defect?

According to this observation, the clinical evaluation of the patients included the performance of an electrocardiogram and an echocardiogram.This information was included in Results section.

4. The author claimed reduced penetrance in case 1. Previously, is there any evidence of reduced penetrance in the EVC2 causing Weyers acrofacial dysostosis? If yes, please mention it in the discussion.

According to this observation, we reanalyzed the WES results and we found that patient 1 had another variant in exon 4 in the donor splicing site, this mutation was inherited by her mother. The reanalyzed results show us that the correct diagnosis is Ellis-van Creveld syndrome, for this reason, as it is an autosomal recessive condition, the parents are only carriers of the variants, so they do not present any clinical manifestations. We made the necessary changes in all the corresponding sections.

5. The third discussion needs to improve and there should be some comparison or relationship of current findings in Case 1 with previous studies to support the results.

According to this observation, the reanalyzed results show us that the correct diagnosis is Ellis-van Creveld syndrome, for this reason, as it is an autosomal recessive condition, the parents are only carriers of the variants. We made the necessary changes in all the corresponding sections.

6. Please check the frequency of the new variants in other databases as well e.g Bravo, all of us, GME.

According to this observation, we reviewed the variant frequencies in the databases sugsested. The frequencies found in these databases are also too low in all populations, mainly in the Latino population. According to the ACMG, this is one of the criteria for considering the detected variants as pathogenic and/or probably pathogenic.

7. Spacing between the words and paragraph needs to double check.

According to this observation, the suggested correction was made.

8. The nomenclature of mutation should follow the same format as HGSV.

According to this observation, the suggested correction was made.

9. In line 319 correct the spelling of the abbreviation ACMG.  

According to this observation, the suggested correction was made.

10. Figure 1 needs to modify, the description is not well formatted.

According to this observation, the suggested correction was made.

11. The protein modeling picture should be larger so can easily visualize.

According to this observation, the suggested correction was made.

12. The web links of databases that are added in the results section can be added after the conclusion under the heading of web sources.

According to this observation, the suggested correction was made. We added the databases links after conclusion under the heading of web sources.

Round 2

Reviewer 1 Report

Overall, the manuscript continues to have many weak points, both as regards the methodology and a confused presentation of the results, and as regards the stylistic aspects; there are redundant sentences on one side and absence of information on the other. It appears also unnecessary the inclusion of a so long description of all the mutations already detected in EVC2 (lines 419-435) with confused conclusion about the concept of genetic heterogeneity. The result is an unclear manuscript, not well organized and difficult to read.

I suggest again to doing a new submission presenting the study as a case report with a simple description of the clinical data and genetic results, so giving it a better focus.

Some minor issues:

In the ‘Materials and Methods’ section there is still a lack of technical information on how the WES was carried out. Moreover, the ‘Results section’ contains unnecessary information not in line with what a results section should contain (lines 240-260).

Also, the figures are a bit confusing without correspondence between results descriptions and electropherograms.

The c.273dup is still listed as ‘novel variant’ (line 30) despite already reported and published.